# Therapeutic Potential of Mesenchymal Stem Cells versus Omega n − 3 Polyunsaturated Fatty Acids on Gentamicin-Induced Cardiac Degeneration

**DOI:** 10.3390/pharmaceutics14071322

**Published:** 2022-06-22

**Authors:** Fatma Y. Meligy, Hanan Sharaf El-Deen Mohammed, Tarek M. Mostafa, Mohamed M. Elfiky, Israa El-Sayed Mohamed Ashry, Ahmed M. Abd-Eldayem, Nermin I. Rizk, Dina Sabry, Eman S. H. Abd Allah, Salwa Fares Ahmed

**Affiliations:** 1Histology and Cell Biology Department, Faculty of Medicine, Assiut University, Assiut 71526, Egypt; fares.s@aun.edu.eg; 2Internal Medicine_Critical Care Department, Faculty of Medicine, Assiut University, Assiut 71526, Egypt; dr_hanansharaf@yahoo.com; 3Anatomy and Embryology Department, Faculty of Medicine, Assiut University, Assiut 71526, Egypt; drtarekmostafahamdan@gmail.com; 4Anatomy and Embryology Department, Faculty of Medicine, Menoufia University, Menoufia 32952, Egypt; mohamed.elfiky@med.menofia.edu.eg; 5Pharmacology Department, Faculty of Medicine, Assiut University, Assiut 71526, Egypt; esraaashry@aun.edu.eg (I.E.-S.M.A.); dr.ahmed2016@aun.edu.eg (A.M.A.-E.); 6Clinical Physiology Department, Faculty of Medicine, Menoufia University, Menoufia 32952, Egypt; nermin.rizk@yahoo.com; 7Medical Biochemistry and Molecular Biology Department, Faculty of Medicine, Badr University in Cairo, Cairo 11829, Egypt; dinasabry@kasralainy.edu.eg; 8Medical Biochemistry and Molecular Biology Department, Faculty of Medicine, Cairo University, Cairo 11562, Egypt; 9Medical Physiology Department, Faculty of Medicine, Assiut University, Assiut 71526, Egypt; eman_sayyed2@yahoo.com or

**Keywords:** cardiac degeneration, mesenchymal stem cells, n − 3 polyunsaturated fatty acids, PCNA, caspase-3, catalase, MDA

## Abstract

This study compared the cardioprotective action of mesenchymal stem cells (MSCs) and PUFAs in a rat model of gentamicin (GM)-induced cardiac degeneration. Male Wistar albino rats were randomized into four groups of eight rats each: group I (control group), group II (gentamicin-treated rats receiving gentamicin intraperitoneally (IP) at dose of 100 mg/kg/day for 10 consecutive days), group III (gentamicin and PUFA group receiving gentamicin IP at dose of 100 mg/kg/day for 10 consecutive days followed by PUFAs at a dose of 100 mg/kg/day for 4 weeks), and group IV (gentamicin and MSC group receiving gentamicin IP at dose of 100 mg/kg/day followed by a single dose of MSCs (1 × 10^6^)/rat IP). Cardiac histopathology was evaluated via light and electron microscopy. Immunohistochemical detection of proliferating cell nuclear antigen (PCNA), caspase-3 (apoptosis), Bcl2, and Bax expression was performed. Moreover, cardiac malonaldehyde (MDA) content, catalase activity, and oxidative stress parameters were biochemically evaluated. Light and electron microscopy showed that both MSCs and PUFAs had ameliorative effects. Their actions were mediated by upregulating PCNA expression, downregulating caspase-3 expression, mitigating cardiac MDA content, catalase activity, and oxidative stress parameters. MSCs and PUFAs had ameliorative effects against gentamicin-induced cardiac degeneration, with MSCs showing higher efficacy compared to PUFAs.

## 1. Introduction

Gentamicin is among many aminoglycoside antibiotics effective against severe life-threatening Gram-negative bacterial infection. Implantable gentamicin–collagen sponges, a novel treatment approach, significantly ameliorates the risk of sternal wound infection after cardiac surgery [1]. Gentamicin offers a good local infection prophylaxis in heart transplant patients [2], valvular disease, infective endocarditis [3], heart valve allograft [4], and preoperative neonates with congenital heart diseases [5].

However, gentamicin exhibits several adverse effects, including nephrotoxicity [6], ototoxicity [7], and cardiac toxicity [8]. Evidence has shown that alterations in cardiac protein expression, α-enolase, and caveolin in the rats’ heart mediate gentamicin-induced cardiac effects [8]. Specifically, gentamicin-induced cardiac injury has been thoroughly investigated considering that the drug impairs free-radical defense systems, causing peroxidation and cellular abnormalities within the heart [9]. Therefore, gentamicin-induced cardiac toxicity could be a representative model of ROS-mediated myocardial injury, such as that observed in ischemia and reperfusion processes and pathological cardiac hypertrophy and failure [10].

Cardiovascular (CV) events represent the primary cause of morbidity and mortality worldwide [11]. Remodeling is a dynamic change that acts as an adaptive or compensatory response following pathological conditions, such as acute myocardial infarction (AMI) [12,13,14]. It usually affects the architecture of the left ventricle and involves rapid hypertrophy of viable myocytes to increase the cardiac mass [15].

Apoptosis, the most morphologically researched process of cell death, can be separated into intrinsic and extrinsic types according to the underlying processes. Microenvironmental problems promote intrinsic apoptosis [16,17,18]. Proapoptotic members of the B cell lymphoma-2 (Bcl2) family, such as Bax, Bak, and BH3-only protein, principally govern intrinsic cell death through their impact on mitochondria. Bcl2 promotes Bax/Bak translocation into mitochondria, causing mitochondrial membrane permeabilization and caspase cascade events [19].

Apoptosis of cardiomyocytes is a well-known critical mechanism in the progression of ischemia [20]. Previous research has found that aberrant Bcl-2 expression plays a key role in cardiomyocyte apoptosis and regulation of myocardial ischemia–reperfusion injury in MI/RI, given that its release rate significantly influences cardiomyocyte apoptosis and contractility [21,22].

Unfortunately, the heart’s ability for endogenous repair of lost cardiomyocytes is insufficient, resulting in progressive heart failure (HF) [23]. Currently available cardiac therapeutic modalities, either invasive or noninvasive, have failed to efficiently compensate for the damaged cardiac tissue [24]. Some novel approaches, including stem cell (SC)-based remedies, have been developed in an attempt to restore cardiac structure and function [25].

Mesenchymal stem cells (MSCs) can be acquired from a variety of adult tissues (e.g., peripheral blood, bone marrow, and adipose tissue) and neonatal tissues (umbilical cord, cord blood, amnion, and placenta) [26]. Nevertheless, bone marrow is recognized as one of the most promising primary sources of MSCs [27].

Transplantation of bone marrow cells induces differentiation of scarred cardiac cells into cardiomyocytes, restoring cardiac function [28]. Given their tremendous ability to proliferate [29,30], decrease infarct size [31], and modify the milieu of injured cardiac tissue to upregulate VEGF [32], bone marrow-derived (BM)-MSCs have shown promising results in cardiac repair and have also been investigated in the context of cardiac stem-cell differentiation [33]. Cai et al. [34] were the first to show that MSCs might be used to treat isopreterenol-induced cardiac hypertrophy. As such, the combined action of BM-MSCs in conjunction with ECM that binds to basic fibroblast growth factor has been shown to improve left-ventricular function [35].

BM-MSCs transdifferentiate of into cardiomyocytes, which express heart-specific transcription factors and produce spontaneously beating cardiac progenitors [36]. These progenitors help repair infarcted hearts and enhance overall heart function. The condition was created using a cryo-injury approach to implant bone marrow MSC/silk fibroin/hyaluronic acid (BMSC/SH) into rat hearts with myocardial infarction [37]. After transplanting BM-MSCs into MI-induced rat hearts, the BM-MSC-treated hearts showed better fractional shortening compared to any of the other models, suggesting a new potential therapeutic strategy for post-infarcted heart failure [38].

Epidemiological studies have shown low incidence of ischemic heart diseases in populations consuming diets rich in fish oil (e.g., Greenland Eskimos and Japanese) compared to European and North American populations [39,40]. The active components responsible for the beneficial effects of the Mediterranean diet are omega-3 polyunsaturated fatty acids comprising docosahexaenoic acid (DHA, C22:6 n − 3) and eicosapentaenoic acid (EPA, C20:5 n − 3) [41]. Studies have shown that polyunsaturated fatty acids (PUFAs) exert cardioprotective effects given that they reduce the risk of CV disorders, including hypertension, cardiac arrhythmias, atherosclerosis, acute myocardial infarction, and sudden cardiac death [42,43]. Their effects have been mainly associated with their potent triacylglycerol-lowering effects in both normolipidemic and hyperlipidemic subjects [42].

The present study examined and compared the cardioprotective effect of MSCs and PUFAs against gentamicin-induced cardiac toxicity in Wistar albino rats by evaluating cardiac histopathology, apoptotis, and the expression of proliferating markers and oxidative stress parameters.

## 2. Materials and Methods

### 2.1. Animals

A total of 32 adult male Wistar albino (3 months old) rats weighing 200 g were used herein and randomized into four groups (*n* = 8/group). Our study protocol was approved by the Medical Ethics Committee, Faculty of Medicine, Assiut University (local approval number: 17200549). Rats were allowed ad libitum access to food and water and kept in stainless-steel cages at the Faculty of Medicine animal house under room temperature and a normal day–night cycle.

### 2.2. Experimental Design

The four groups were as follows: group I, control group receiving saline intraperitoneally (IP); group II, gentamicin-treated group receiving 80 mg ampoules of gentamicin (Memphis Pharm. & Chemical Ind., Cairo, Egypt) IP at a dose of 100 mg/kg/day for 10 consecutive days [44]; group III, PUFA group receiving 80 mg ampoules of gentamicin (Memphis Pharm. & Chemical Ind., Cairo, Egypt) IP at a dose of 100 mg/kg/day for 10 consecutive days [44] followed by PUFAs (Doppelherz, Flensburg, Germany) at a dose of 100 mg/kg/day [45] via the intragastric route using an oral gavage tube for 4 weeks; group IV, MSC group receiving 80 mg ampoules of gentamicin (Memphis Pharm. & Chemical Ind., Cairo, Egypt) IP at dose of 100 mg/kg/day for 10 consecutive days [44] followed by a single dose of MSC at passage two (1 × 10^6^)/rat IP.

### 2.3. Isolation, Culture, Immunophenotyping, and Differentiation of Bone Marrow-Derived Mesenchymal Stem Cells

The Mesenchymal and Tissue Stem Cell Committee of the International Society for Cellular Therapy established some criteria for MSCs, including the ability of MSCs to proliferate in the form of plastic adherent cells in standard culture conditions, the expression of CD90, CD105, CD44, and CD29; and the lack of expression of CD45, CD34, CD31, CD14 or CD11b, CD79a, and CD19, in addition to the capacity to differentiate into multiple cell lineages in vitro [46].

For isolation of BM-MSCs, three male Wistar albino rats (3 weeks old, weighing 25–30 g) were euthanized and placed in a vertical laminar airflow hood where the humerus and femur were dissected under aseptic conditions, cleaned from all attached muscles and all connective tissues, and assembled on ice in 10 mL of αMEM isolation medium (IM) (ThermoFisher Scientific, Paisley, UK, Cat. No. 22571038), supplemented with 10% antibiotic–antimycotic (Sigma-Aldrich, Merck KGaA, Darmstadt, Germany Cat. no. 15240062). The two ends of the humerus and femur were cut. BM-MScs were obtained by inserting a needle into the medullary cavity and flushing the bone marrow into a 15 mL centrifuge tube with αMEM (Sigma-Aldrich. Cat. no. CLS430055). To concentrate the cells, the flushed bone marrow was centrifuged for 5 min at 1000 rpm. The supernatant was thrown away. Then, 10 mL of complete medium was pipetted into the tube and thoroughly mixed. The complete medium comprised αMEM with 20% heat-inactivated fetal bovine serum (FBS, Thermo-Fisher Scientific, Cat. no. 26140087) and 1% antibiotic–antimycotic (Sigma-Aldrich, Cat. no. 15240062), filtered through a 70 μm nylon cell strainer (Sigma-Aldrich, cat no. CLS431751) into T-75 tissue culture flasks (Greiner Bio-One International, Kremsmünster, Austria, Cat. no. 156472). Cells were incubated in the Co_2_ incubator at 37 °C and 5% CO_2_. Nonadherent cells were removed by changing the medium. The medium was changed every 2 days. The cells reached the 80% confluence on the seventh day of culture [47]. For immunophenotyping of MSCs, the cultured cells of passage two were trypsinized using 1× trypsin EDTA solution, 0.25% trypsin, and 0.02% EDTA (Sigma-Aldrich, Cat. no. 59428C). The cells were then washed in (PBS) phosphate-buffered saline (Sigma-Aldrich, Cat. no.: 10010023). Next, they were incubated with CD105 (Santa Cruz biotech, Santa Cruz, CA, USA, Cat. no.: sc-71042), CD29 (Santa Cruz biotech, Santa Cruz, CA, USA, Cat. no.: sc-9970), CD44 (Santa Cruz biotech, Santa Cruz, CA, USA, Cat. no.: sc-7297), CD90 (Santa Cruz biotech, Santa Cruz, CA, USA, Cat. no.: sc-53116), CD34 (Santa Cruz biotech, Santa Cruz, CA, USA, Cat. no.: sc-7324), and CD45 (Santa Cruz biotech, Santa Cruz, CA, USA, Cat. no.: sc-1178) primary antibodies in 1% bovine serum for 30 min. (Table 1). Then, the cells were washed using phosphate-buffered saline (Sigma-Aldrich Cat. no.: 10010023) and centrifuged for 5 min at 1000 rpm. The cells were incubated with the secondary antibody (Alexa Fluor 647; Santa Cruz biotech, Santa Cruz, CA, USA, Cat. no.: sc-24636) for 30 min, washed twice for 5 min each, and then subjected to an FACS cell analyzer [48].

For differentiation of isolated BM-MSCs, Cells of the third passage were planted into a 24-well culture plate at a concentration of 2 × 10^6^ cells/well after reaching 70–80% confluence.

For adipogenic differentiation, adipogenic differentiation medium (Stempro Sigma-Aldrich Cat. No.: A1007001) was changed twice weekly and was maintained for 4 weeks. Oil Red O staining (Sigma-Aldrich Cat. No.: MAK194) was used to confirm positive adipogenic differentiation, which appeared as lipid droplets inside the cell. For chondrogenic differentiation, chondrogenic differentiation medium (Stempro, Sigma-Aldrich Cat. No.: A1007101) was maintained for 18 days with twice weekly addition of medium. Chondrogenic differentiation was assessed by Alcian Blue staining (Sciencell Cat. no.: 8378) [49]. For osteogenic differentiation, Osteogenic differentiation medium (Stempro, Sigma-Aldrich, Cat. No.: A1007201) was changed weekly and maintained up to 3 weeks. Calcium deposition was evaluated by Alizarin Red staining (Sciencell, Cat. no.: 8678), to assess the differentiation potential of cells [49].

### 2.4. Determination of Cardiac Malonaldehyde Content

The cardiac content of malonaldehyde (MDA), the end product of lipid peroxidation, was assessed according to the method described by Wasowicz et al. [50], which was based on the reactivity of thiobarbituric acid (TBA). In 10 mL glass tubes containing 1 mL of distilled water, 50 of μL of homogenate or MDA working standard solution was added. The samples were placed in a water bath and heated for 1 h at 95–100 °C after adding 1 mL of a solution containing 29 mmol/L TBA in acetic acid (pH of the reaction mixture, 2.4–2.6) and mixing. After cooling the samples, 25 L of 5 mol/L HCl (final pH 1.6–1.7) was added, and the reaction mixture was extracted with 3.5 mL of *n*-butanol after 5 min of agitation. The butanol phase was separated by centrifugation at 1500× *g* for 10 min. A Perkinelmer fluorometer was used to measure the fluorescence of the butanol extract (Model LS50B, Perkin-Elmer, Buckinghamshire, UK) at wavelengths of 525 and 547 nm for excitation and emission at wavelengths of 525 and 547 nm for excitation and emission, respectively.

### 2.5. Determination of Cardiac Catalase Activity

Cardiac catalase activity was measured as described by Goth [51]. At the end of the experiment, the hearts were rapidly excised, bathed in ice-cold PBS (pH 7.4), homogenized in 50 mM Tris buffer (pH 7.4) containing 400 mM NaCl and 0.5% Triton X-100 using a Glas-Col Homogenizer AQ5, and centrifuged at 10,000 rpm for 10 min by ultracentrifugation (Hettich EBA 12) at approximately 4 °C. The supernatants were collected and kept at –20 °C until use. The enzymatic process was stopped by adding 1 mL of ammonium molybdate to the homogenate incubated in H_2_O_2_ substrate. The intensity of the yellow complex produced by molybdate and H_2_O_2_ was detected at 405 nm.

### 2.6. Histological Studies

Rats from all groups were anesthetized with thiopental sodium (50 mg/kg, IP). Rats were then perfused with saline until clear flow was observed, after which they were perfused with 10% formalin. After perfusion, heart specimens were removed carefully and then refixed in 10% neutral-buffered formalin (pH 7.2), dehydrated, cleared, and embedded in paraffin. Coronal paraffin sections were cut to a size of 4–6 μm and stained with hematoxylin and eosin (H&E) for general histological study [52]. Masson’s trichrome staining was used to determine changes in collagen distribution during the healing and reconstruction process in damaged heart tissue [52].

### 2.7. Immunohistochemical Studies

Caspase-3 (a rabbit polyclonal antibody purchased from AB clonal Technology, Woburn, MA, USA, Cat. No.: A0214), proliferating cell nuclear antigen (PCNA; a mouse monoclonal antibody, purchased from Novus Biologicals, Centennial, CO, USA, Cat. No.: NB500-106SS), Bax (rabbit polyclonal antibody purchased from Biospes, Chongqing, China, Cat. No.: YPA2175), and Bcl2 (Rabbit polyclonal antibody purchased from Biospes, China, Cat. No.: YPA2275) were used for detecting apoptosis. The data related to the antibodies used in immunohistochemical studies are found in Table 2 Deparaffinized paraffin-embedded sections (5 m) were rehydrated in alcohol, boiled in 10 μM citrate buffer (pH 6.0) for 15 min, and then cooled at room temperature for 20 min. All antibodies were used at a dilution of 1:100 for 30 min. Processing of sections was conducted according to the manufacturer’s instructions using the universal kit (EcnoTek HRP Anti-Polyvalent, DAB; ScyTek Laboratories, Inc., 205 South 600 West, Logan, UT, USA). After the reaction was completed, Mayer’s hematoxylin was used to counterstain the area, which was then dehydrated and covered with DPX [53].

### 2.8. Quantitative Analyses

Quantification was performed on a minimum five nonoverlapping fields in three serial immunostained sections from three different animals. Image J software (version 1.53i) was used for quantitative analysis of PCNA, caspase-3, Bcl2, and Bax. To evaluate variations in immunoreactivity for PCNA, caspase-3, Bcl2, and Bax among different study groups, the DAB signal was quantified using Image J software. From the “analyze menu”, we selected “set measurement” and checked “area”, “max. gray value”, and “mean gray value” from the resulting popup window, which were then measured. The optical density was determined using the equation OD = log (max. gray intensity/mean gray intensity) to determine the degree of immunoreactivity (darkness) of the stained cells using the DAB signal.

For measuring the area of the green collagen fibers between cardiac myocytes in slides stained with Masson’s trichrome, we chose color from the analyze menu and then color deconvolution and threshold were adjusted until the green color of collagen appeared. Finally, we obtained the percentage of collagen area.

### 2.9. Electron Microscopy Studies

Heart tissues were taken as full-thickness slices from the left ventricle’s wall for electron microscopy.

This midventricular slice was fixed in 4% glutaraldehyde for 4 h at 4 °C [54]. The glutaraldehyde-fixed specimens were cut into semithin sections (0.5–1 m thick) and stained with toluidine blue. Using the transmission electron microscope, ultrathin sections (50–80 nm) were cut from selected portions of semithin sections, contrasted with uranyl acetate and lead citrate [55], analyzed using an electron microscope (JEOL, JEM-100CXII Akishima, Tokyo, Japan), and photographed at 80 kV in the Electron Microscope Unit of Assiut University.

### 2.10. Statistical Analysis

GraphPad Prism Version 7 was used for the statistical analysis. Nonparametric statistics were used. The Kruskal–Wallis test was used to examine the data, followed by Dunn’s test. The median and interquartile range were used to express the findings. Differences were considered significant at *p* < 0.05.

## 3. Results

### 3.1. Characterization of BM-MSCs

Cultured isolated BM-MSCs had a fibroblast-like shape with elliptical nucleus and fusiform shape (Figure 1a). After being cultured in specialized induction medium, BM-MSCs underwent trilineage differentiation. The ability of BM-MSCs to differentiate was discovered using a specific staining approach. Special staining with Alizarin Red, Oil Red O, and Alcian Blue revealed the osteogenic, adipogenic, and chondrogenic differentiation potential, respectively. Alizarin Red S staining was used to show calcium deposits in the matrix. This histological staining is based on alizarin red’s ability to specifically stain the calcium-containing matrix, and its positive appearance is thought to indicate bone matrix deposition, thus validating osteogenic differentiation. The formation of proteoglycans by rat MSCs was confirmed by Alcian Blue staining, revealing the differentiation to chondrocytes. Oil Red O staining revealed intracellular lipid-filled droplets that stained deep red, indicating that rat MSCs had the potential to differentiate into adipocytes. Isolated BM-MSCs were positive for CD105, CD29, CD44, and CD90 but negative for CD34 and CD45 (Figure 1e).

### 3.2. Effects of PUFAs and MSCs on Gentamicin-Induced Alterations in Cardiac Catalase Activity and Cardiac MDA Content

Cardiac MDA content was significantly enhanced in the gentamicin-induced cardiac degeneration model (*p* = 0.0007). Treatment with MSCs reduced MDA expression significantly (*p* = 0.0068). An insignificant change in MDA expression was observed in the PUFA-treated group (*p* > 0.9999) (Figure 2). Cardiac catalase activity was significantly reduced in the gentamicin-induced cardiac toxicity degeneration model (*p* = 0.0457). Treatment with either MSCs or PUFAs significantly enhanced catalase activity (*p* = 0.0003; *p* = 0.003, respectively) (Figure 2).

### 3.3. Histological Results

#### Light Microscopic Examination

Among heart tissue sections stained with H&E, the control group showed normal histological structure of the myocardium. In longitudinally cut sections, the myocytes appeared branched and striated. The cytoplasm of cardiomyocytes was acidophilic with oval nuclei that were centrally located (Figure 3a). The gentamicin-treated group revealed cytoplasmic loss and fragmentation in some parts of cardiomyocytes fibers with loss of their striations. Nuclei of the cardiomyocytes showed deformation in sizes and shapes, while others appeared pyknotic. Interstitial hemorrhage could be detected (Figure 3b). The gentamicin- and PUFA-treated group showed an apparently regular arrangement of cardiac muscle fibers (Figure 3c). The gentamicin and MSC-treated group exhibited a nearly regular arrangement of cardiac muscle fibers (Figure 3d). In Masson’s trichrome-stained sections, scanty collagen fibers between cardiac muscle fibers were detected in control group (Figure 4a,e). The gentamicin-treated group showed significantly more collagen fibers between the cardiac muscle fibers (Figure 4b,e) compared to the control group (*p* < 0.0001). The gentamicin- and PUFA-treated group revealed fewer collagen fibers between cardiac muscle fibers compared to group II (*p* = 0.528) (Figure 4c,e). A minimal amount of collagen fibers between cardiac muscle fibers were detected in the gentamicin- and MSC-treated group (*p* = 0.004) (Figure 4d,e). Semithin sections appeared as basophilic longitudinal bundles with oval, vesicular, and centrally located nuclei. Transverse striations were markedly observed (Figure 5a). The gentamicin-treated group revealed fragmentation of myocytes and cytoplasmic fibrinolysis, which induced a loss of and decrease in component fibrils. Myofiber striations were absent in some places (Figure 5b). In the gentamicin- and PUFA-treated group, cardiac myocytes nearly retained their regular arrangements with the appearance of striations (Figure 5c). The gentamicin- and MSC-treated group revealed cardiac myocytes that had their regular appearance and myofibrillar arrangements and contained a vesicular central nucleus, while transverse striations became prominent (Figure 5d).

### 3.4. Effect of PUFAs and MSCs on Gentamicin-Induced Alterations in Cardiac Expression of Caspase-3

To investigate whether PUFA and MSC treatment influences gentamicin-induced apoptosis, we determined caspase-3 expression, a cardiac apoptotic parameter, using immunohistochemistry. In controls, cardiac caspase-3 expression was minimal (Figure 6a). In contrast, injection of gentamicin increased cardiac caspase-3 expression markedly (Figure 6b). However, PUFA treatment promoted lower caspase-3 expression in the cardiac myocytes (Figure 6c) compared to animals treated with gentamicin alone. Moreover, MSC treatment decreased caspase-3 expression in the cardiac myocytes (Figure 6d). After quantifying the intensity of caspase-3 expression via immunohistochemistry, we found that rats treated with gentamicin had a significantly greater increase in caspase-3 expression compared to control animals (*p* < 0.0001). However, MSC treatment promoted significantly lower caspase-3 expression compared to gentamicin-treated animals (*p* = 0.0039), while PUFAs resulted in an insignificant reduction (*p* = 0.528) (Figure 6e).

### 3.5. Effect of PUFAs and MSCs on Gentamicin-Induced Alterations in the Cardiac Expression of PCNA

To examine whether gentamicin affects cellular proliferation and whether PUFA and MSC treatment impacts cell proliferation, we evaluated the expression of PCNA, a cellular proliferation marker in cardiac tissue, using immunohistochemistry. Accordingly, control rats showed moderate PCNA expression in the cardiac myocytes (Figure 7a). In contrast, gentamicin injection markedly decreased PCNA expression in cardiac myocytes (Figure 7b). However, PUFA treatment promoted significantly greater PCNA expression in cardiac myocytes (Figure 7c) compared to gentamicin alone. Similarly, MSC treatment significantly increased PCNA expression in cardiac myocytes (Figure 7d). Quantitative analysis of PCNA expression via immunohistochemistry showed a significant decrease in PCNA expression among rats treated with gentamicin compared to control animals (*p* < 0.0001). However, MSC treatment promoted a significantly greater PCNA expression compared to gentamicin treatment (*p* = 0.004), with an insignificant effect of PUFAs (*p* = 0.381) (Figure 7e).

#### Effect of PUFAs and MSCs on Gentamicin-Induced Alterations in the Cardiac Expression of Bax/Bcl2

Immunohistochemical investigation of Bcl2 revealed high expression in the control group (Figure 8a). In GM-treated animals, very weak expression was detected (Figure 8b). Administration of PUFAs slightly increased Bcl2 expression (Figure 8c), while BM-MSC treatment reversed the immunoreactivity of Bcl2 considerably when compared to the GM group (Figure 8d). Quantitative analysis of Bcl2 expression via immunohistochemistry showed a significant decrease in Bcl2 expression among rats treated with gentamicin compared to control animals (*p* < 0.0001) However, in PUFA-treated rats, the increase was not statistically significant (*p* = 0.5280). MSC treatment promoted a significantly greater Bcl2 expression compared to gentamicin treatment (*p* = 0.0037) (Figure 8i). In contrast, the administration of GM greatly elevated Bax expression (Figure 8f) in comparison to control animals (Figure 8e), indicating the enhanced apoptotic process. PUFA administration produced a weak impact on Bax expression when compared to GM animals (Figure 8g) (*p* = 0.5287). BM-MSC treatment attenuated Bax immunoreactivity (Figure 8h). The effect of MSC therapy was dramatically significant versus the GM group (*p* = 0.0039). Expression in the GM-treated group was statistically significant in comparison to control group (*p* < 0.0001). Quantitative analysis of Bax expression via immunohistochemistry is shown in Figure 8i.

### 3.6. Electron Microscopy Results

Control rat cardiac myocytes contained oval and centrally placed nuclei with finely scattered chromatin, which were surrounded by an intact nuclear membrane, according to electron microscopy (Figure 9a). The sarcolemma limiting cardiac myocytes displayed invaginations at the Z-lines to generate transverse tubules (T tubules) (Figure 9b). Mitochondria were observed between the myofibrils and had a spherical or elongated shape, were abundant, and were arranged regularly in rows between the myofibrils. The myofibrils were grouped in sarcomeres between Z lines, with a dark A band in the center and a light I band on the edges. A light H zone could also be observed in the dark A band center (Figure 9a–c). In gentamicin-treated rats, with lysis of the myofibrils and enlarged interfibrillar gaps, cardiac myocytes were fragmented and disorganized (Figure 9a). In the nuclei, chromatin condensation in the nucleoplasm was detected as an atypical chromatin pattern (Figure 9b). These changes caused disorganization and disarrangement of Z lines and H lines. The mitochondria had disrupted cristae (Figure 9c). In the gentamicin- and PUFA-treated group, myofibrils exhibited better reconstruction and reorganization compared to those in group II, although some loss was still observed (Figure 9a). The chromatin pattern of vesicular nuclei was observed to be normal (Figure 9b). Although rearrangement and continuation of Z lines and H lines were observed, some areas had a discontinued Z line. Regular cell membranes and their invagination could be seen at the site of the T tubules. Mitochondria in group III were better organized in between myofibrils compared to those in group II (Figure 9c). The ultrastructural sections of the gentamicin- and MSC-treated group showed reconstruction and reorganization of myofibrils, with vesicular nuclei showing normal chromatin patterns and regular cell membranes (Figure 9a,b). We observed Z- and H-line rearrangement and continuation, as well as invagination at the T tubule site. Mitochondria in group III were better organized between myofibrils compared to those in group II (Figure 9c).

## 4. Discussion

The current study highlights the toxic potential of gentamicin in cardiac tissue. Accordingly, our results revealed that gentamicin-treated rats had cytoplasmic loss and fragmentation of cardiac muscle fibers with loss of their striations. Cardiac myocytes were fragmented and disorganized, with lysis of the myofibrils and expansion of interfibrillar gaps. In the nuclei, chromatin condensation in the nucleoplasm was detected as an atypical chromatin pattern. These changes caused disorganization and disarrangement of Z lines and H lines. The aforementioned findings are consistent with clear apoptotic changes. The oxidative stress scenario of the effects of gentamicin offers a good explanation for our histological findings, which are further supported by our biomedical measures of catalase, an antioxidant enzyme, and MDA, a lipid peroxidation product and an important marker of oxidative stress. The present study showed that gentamicin-treated rats had significantly reduced catalase activity but significantly enhanced MDA levels. Treatment with either MSCs or PUFAs significantly enhanced catalase activity and reduced MDA levels, with the MSC group showing greater enhancement in catalase activity and a reduction in MDA levels compared to the PUFA group. Actually, an insignificant change in MDA expression was observed in the PUFA-treated group.

Catalase stopped the progression of overt heart failure and the cellular hallmarks of unfavorable remodeling (myocyte hypertrophy, myocyte death, and interstitial fibrosis). Catalase overexpression in the heart inhibits the progression of overt heart failure by involving hydrogen peroxide-dependent and -independent phases of myocardial remodeling [56].

Consistent with our study, Randjelovic et al. [57] reported that gentamicin-induced nephrotoxicity was associated with the development of superoxide anion, hydrogen peroxide, and hydroxyl radical reactive oxygen species (ROS) from renal cortical mitochondria, followed by an increase in lipid peroxidation and a reduction in antioxidant enzymes, thereby elevating renal oxidative stress.

Our findings demonstrated that the gentamicin-treated group had higher Bax expression than the control group, but the MSC-treated groups had lower Bax expression than the gentamicin-treated group. Similarly, overexpression of the Bax gene and caspase-3 was correlated with gentamycin-induced histopathological changes in mitochondria, but the level of the Bcl-xL gene was considerably lowered in the kidney [58], liver [59], and sensory hair cells [60].

Proteins in the BCL2 family are continually in interaction with one another, resulting in a variety of cell outcomes [61]. Bax and Bak are mitochondrial proteins that increase membrane permeability in response to apoptotic activation and outer membrane protein oligomerization [62,63]. When Bcl2 interacts with Bax and Bak, it prevents mitochondrial membrane permeabilization and induces cell death [61,64]. Increased Bcl2 expression could make cells more resistant to apoptosis. Bcl2 is a critical protein in the Bcl2 family that plays a role in antiapoptosis and cardiomyocyte survival [65]. In comparison to controls, rats receiving MSC therapy had a higher ventricular (Bcl2)/Bcl2-associated X protein (Bax) ratio and reduced caspase-3 protein expression [66,67]. In ventricular cardiomyocytes, Bcl2 is a key inhibitor of apoptosis, whereas Bax is a proapoptotic protein. As a result, a higher Bcl2/Bax ratio indicates inhibition of cardiac apoptotic pathways [66] and lower Bax expression in cardiomyocyte cells [68]. Low ratios of n − 6/n − 3 PUFAs, on the other hand, were found to dramatically reduce serum inflammatory markers, infarct size in MI/RI rats, the number of cardiomyocytes undergoing apoptosis, and caspase-3, Bcl-2, and Bax expression levels [69].

Enhanced ROS accelerates the permeabilization of the mitochondrial outer membrane by generating proapoptotic Bcl2 superfamily proteins [70]. Moreover, ROS can induce apoptosis by triggering signaling pathways for the initiation of apoptosis or by inhibiting cell protective mechanisms [71]. Myofibrillary degeneration (myocytolysis) and the lack of transverse striations in affected fibers, as observed herein, can be considered a characteristic of distinctive sublethal cardiac muscle cell injury, and it was clearly defined as an ischemic effect on cardiomyocytes by Haschek et al. [72].

Our results suggested that treatment with gentamicin increased the percentage of collagen fibers, which could be attributed to increased ROS production [73]. We noticed less collagen fiber deposition in interstitial tissues following PUFA and BM-MSC treatment. These changes could be due to the antioxidant and anti-inflammatory properties of PUFAs and MSCs, which inhibit lipid peroxidation and removal of free radicals [74]. The anti-inflammatory property of n − 3 PUFAs was mediated by antagonizing the activity of arachidonic acid, suppressing the cell-mediated immune response and leukocytic activity [75]. The antifibrotic effects of omega-3 in pulmonary and cardiac tissues were previously recognized [76], promoting downregulation of profibrogenic genes or changes in the composition of the cell membrane with a subsequent decrease in inflammation and fibrosis [77].

BM-MSC injection decreases collagen fibers by decreasing the expression of profibrogenic factors, including TGF-β and α-SMA, and by controlling the expression of MMPs, α-SMA, TGF-β, TIMP1, and COL1A2 involved in fibrosis [78]. MSCs alleviate pulmonary fibrosis in mice as evidenced by the significant decrease in collagen deposition, lowered expressions of TGF-β1, vimentin, and p-Smad2/3, and increased expression of E-cadherin [79].

PCNA, a key factor in DNA replication and repair pathways, which plays an important role in genome stability, was investigated to determine the underlying molecular mechanisms of the effect of gentamicin and the protective effect of PUFAs and MSCs.

There is a substantial energetic network of PCNA-linked molecules that regulate the activities of proteins required for DNA replication [80].

In the present study, the significantly fewer PCNA immune-stained cells in the gentamicin-treated group compared to those in the control indicated reduced proliferative activity in the former. High levels of ROS have been suggested to cause DNA fragmentation and chromatin crosslinking [81]. In vitro studies have shown that gentamicin inhibits the proliferation of human epidermal keratinocytes, most likely by inhibiting transport RNA [82]. Gentamicin also inhibits cell proliferation in the Corti organ and human osteoblasts [83]. Randjelovic et al. [57] reported that gentamicin boosts the expression of inducible nitric oxide synthase (iNOS) mRNA and NO production in mesangial cells, playing an antiproliferative role in these cells. Conversely, the present work showed that PCNA expression significantly increased in PUFA- and MSC-treated groups. PCNA contributes to BM-MSC proliferation, resistance to apoptosis, and stemness maintenance and significantly decreases with decreased stem cell activity [84]. Al-Bedhawi [85] demonstrated that increased PCNA expression by TMEM121 results in proliferation of the stem/progenitor cells. Omega-3 improved the spermatogenic function in diabetic rats by enhancing antioxidant activity, increasing Bcl2, an antiapoptotic marker, and increasing the expression of PCNA [86].

Regarding immunohistochemistry findings for caspase-3, the current study showed that the gentamicin-treated group had increased caspase-3 expression, which was consistent with the findings of Babaeenezhad et al. [87]. Gentamicin has been shown to boost the production of ROS. Increased ROS in gentamicin-induced acute kidney injury plays a pivotal role in mitochondrial dysfunction, which stimulates the intrinsic pathway of apoptosis [88].

The cardiac myocytes of gentamicin-treated rats were fragmented and disorganized, with lysis of myofibrils, enlarged interfibrillar gaps, and condensation of chromatin in the nucleoplasm. These changes caused disorganization and disarrangement of Z lines and H lines. The mitochondria had disrupted cristae, although rearrangement and better organization of cell organelles and continuation of Z-lines and H-lines were observed in PUVA and MSC groups. Light microscopy has largely been used to characterize morphological alterations associated with apoptosis, such as cell pyknosis, shrinkage, and cellular membrane blebbing. Thus, mitochondriosis, chromatin condensation, cytoplasmatic vacuoles and shrinkage, myofibrillar disarray or lysis, and apoptotic body formation should be employed to diagnose myocardial apoptosis using electron microscopy [89]. During the progression of drug-induced cardiomyocyte apoptosis, electron microscopy consistently revealed apoptosis-related ultrastructural variations in cardiomyocytes, as evidenced by nuclei with chromatin condensation, while mitochondriosis disrupted cytoskeletal structures, cellular fragmentation, and specific cell–cell connections [90], in addition to DNA breakage, cell shrinkage, macrophage phagocytosis, and other noninflammatory effects. Plasma integrity loss, cell disintegration, and engulfment by neighboring cells were also observed [91]. Unlike light microscopic preparations, electron microscopy can detect ultrastructural alterations associated with apoptosis, such as nuclear fragmentation (karyorrhexis) and modest cytoplasmic organelle abnormalities [92]. In cardiomyocytes, three primary apoptosis signaling routes have been identified: (a) endoplasmic reticulum system, (b) mitochondrial pathway, and (c) extrinsic death receptor pathway (e.g., TNF and FAS) [93,94].

With respect to the ultrastructural changes in mitochondria, gentamicin promoted electron density loss and cristae disruption and loss in some mitochondria. Mitochondria are the “powerhouse” of eukaryotic cells, such as cardiomyocytes, which need increased energy supply [95]. Notably, mitochondria occupy nearly 30% of the total cell volume of these cells [96]. Maintenance of effective inter-organelle communication and the “mitochondrial quality control” system is important for maintaining the demand for mitochondrial bioenergetics and metabolic functions [97]. Brown et al. [98] showed that instability in the structure and function of cardiac mitochondria promotes cardiovascular disease pathogenesis. Most of these changes were modified by using PUFAs and BM-MSCs, which is in accordance with Lee et al. [99], who documented that stem cells release regenerative or immunomodulatory factors that can repair local damage, such as heart or brain infarction. In addition, MSCs can guard against tissue damage via paracrine behavior [100].

## 5. Conclusions

In conclusion, PUFAs and BM-MSCs exert ameliorative effects against gentamicin-induced cardiac degeneration. They can protect the myocardium by promoting the differentiation of myocardial cells, enhancing apoptosis resistance, and decreasing oxidative stress, all of which are beneficial to cardiovascular repair. However, our findings showed that BM-MSCs had better efficacy compared to PUFAs. This study provides insight into the consideration of BM-MSC therapy as a novel and promising approach for the treatment of cardiovascular diseases.

## Figures and Tables

**Figure 1 pharmaceutics-14-01322-f001:**
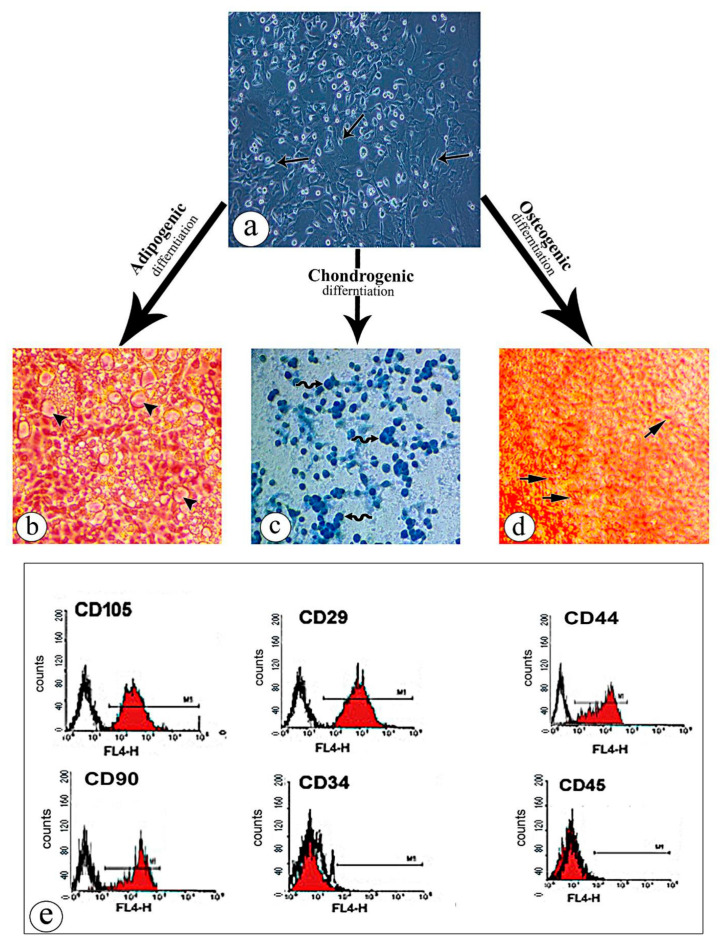
Characterization of bone marrow-derived mesenchymal stem cells: (**a**) the adherent cells appeared fusiform in shape with an elliptical nucleus (arrow); (**b**) BM-MSCs differentiated into adipocytes and stained with Oil Red O exhibited intense cytoplasmic staining, signifying the accumulation of lipid vacuoles (head arrow); (**c**) BM-MSCs differentiated into chondrocytes, exhibiting positive Alcian Blue staining (wavy arrow); (**d**) BM-MSCs differentiated into osteocytes and stained with Alizarin red staining, indicating clusters of calcium depositions (thick arrow); (**e**) FACS analysis of isolated stem cells showing that the bone marrow-derived mesenchymal stem cells were strongly positive for CD105, CD29, CD44, and CD90 markers but negative for CD34 and CD45 markers.

**Figure 2 pharmaceutics-14-01322-f002:**
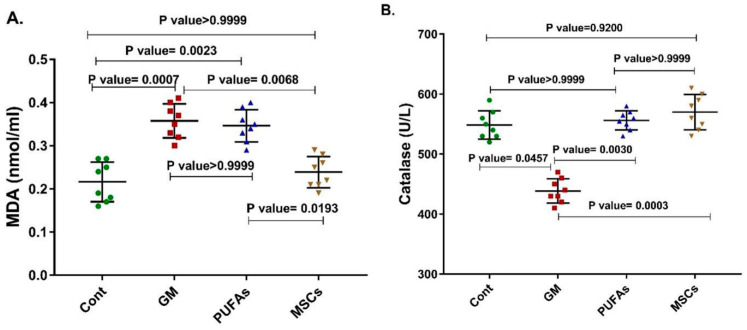
Cardiac MDA content and catalase activity in all experimental groups. (**A**) MDA is a lipid peroxidation product and an important marker of oxidative stress. The MDA content was significantly enhanced in the gentamicin-induced cardiac degeneration model (*p* = 0.0007). Treatment with MSCs reduced MDA expression significantly (*p* = 0.0068). An insignificant change in MDA expression was observed in the PUFA-treated group (*p* > 0.9999). (**B**) Catalase is an antioxidant enzyme. Its activity was significantly reduced in the gentamicin-induced cardiac degeneration model (*p* = 0.0457). Treatment with either MSCs or PUFAs enhanced catalase activity significantly (*p* = 0.0003; *p* = 0.003, respectively). Nonparametric statistics were used. Data were analyzed using Kruskal–Wallis test followed by Dunn’s test. Results were expressed as medians and interquartile ranges. Differences were considered significant at *p* < 0.05.

**Figure 3 pharmaceutics-14-01322-f003:**
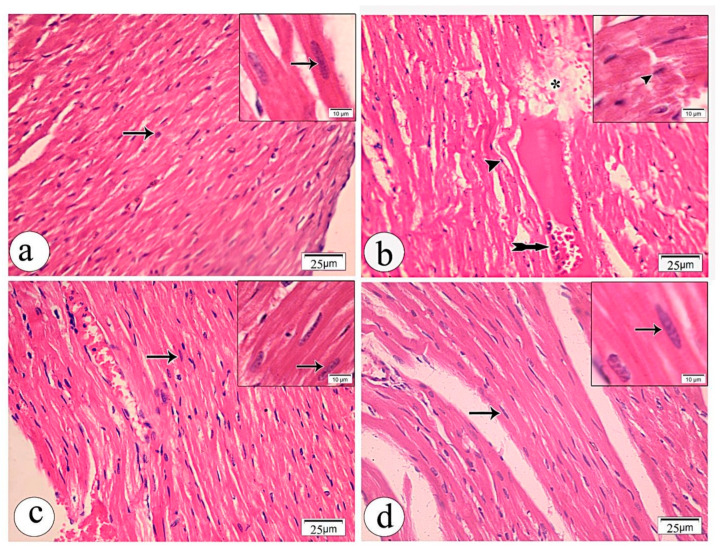
Photomicrographs of heart sections stained by hematoxylin and eosin: (**a**) control group showing the normal histological structure of cardiac myocytes that appeared arranged in a linear array that branched and anastomosed with acidophilic sarcoplasm and oval, centrally located nuclei (arrow); (**b**) gentamycin-treated group showing that most of the cardiac muscle fibers were disorganized and lost the normal striations, while nuclei of the cardiomyocytes showed deformation in sizes and shapes, and others appeared pyknotic (Head arrow), in addition to cytoplasmic loss and fragmentation of cardiac muscle (star) and interstitial hemorrhage (tailed arrow); (**c**) gentamycin- and PUFA-treated group showing regular arrangement of the cardiac muscle fibers with oval, centrally located vesicular nucleus (arrow); (**d**) gentamycin- and MSC-treated group showing regularly arranged cardiac myofibers with oval, rounded nuclei (arrow). Note that inset images of higher magnification were added in all panels photos to show the nucleus.

**Figure 4 pharmaceutics-14-01322-f004:**
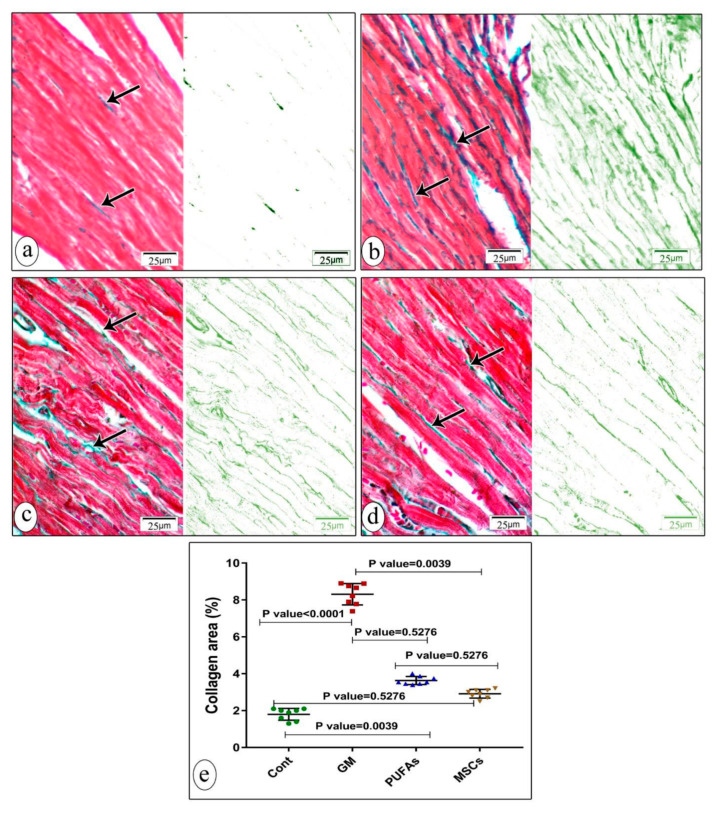
Photomicrographs of sections of heart stained by Masson trichrome: (**a**) control group showing scanty collagen fibers in green color in between the cardiomyocytes which appear dark red in color (arrow); (**b**) gentamycin-treated group showing increased collagen fibers between the cardiomyocytes (arrow); (**c**) gentamycin- and PUFA-treated group showing less serious accumulation than gentamycin group(arrow); (**d**) gentamycin- and MSC-treated group showing reduced collagen accumulation (arrow). (**e**) Statistical analysis of collagen score. Nonparametric statistics were used. Data were analyzed using the Kruskal–Wallis test followed by Dunn’s test. Results are expressed as medians and interquartile ranges. Differences were considered significant at *p* < 0.05.

**Figure 5 pharmaceutics-14-01322-f005:**
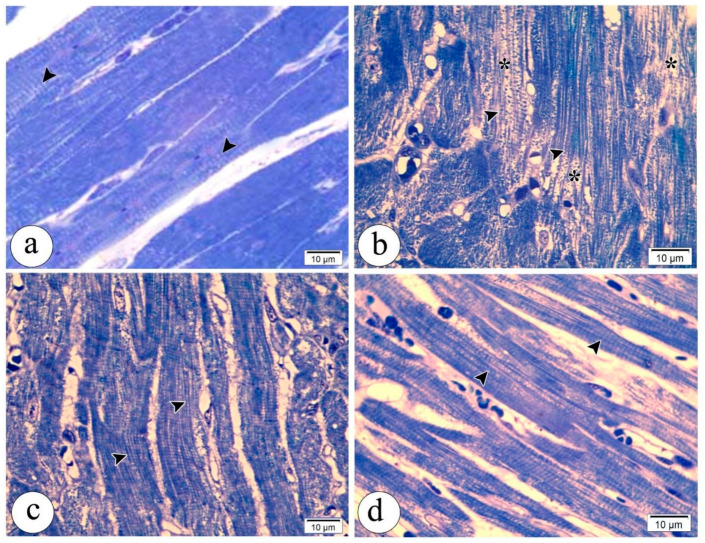
Semithin sections of cardiac muscles showing transverse striations: (**a**) Control group showing cylindrical cardiac muscle fibers and central, oval, and vesicular nuclei, along with visible transverse striations (head arrow); (**b**) gentamycin-treated group showing loss of striation in some parts of the cardiomyocytes( head arrow), along with areas of fibrinolysis (star); (**c**) gentamycin- and PUFA-treated group showing restored normal striation of cardiac muscle fibers but less prominent striations compared with the control group (head arrow); (**d**) gentamycin- and MSC-treated group showing more or less normal cardiac muscle fibers with visible transverse striations (head arrow).

**Figure 6 pharmaceutics-14-01322-f006:**
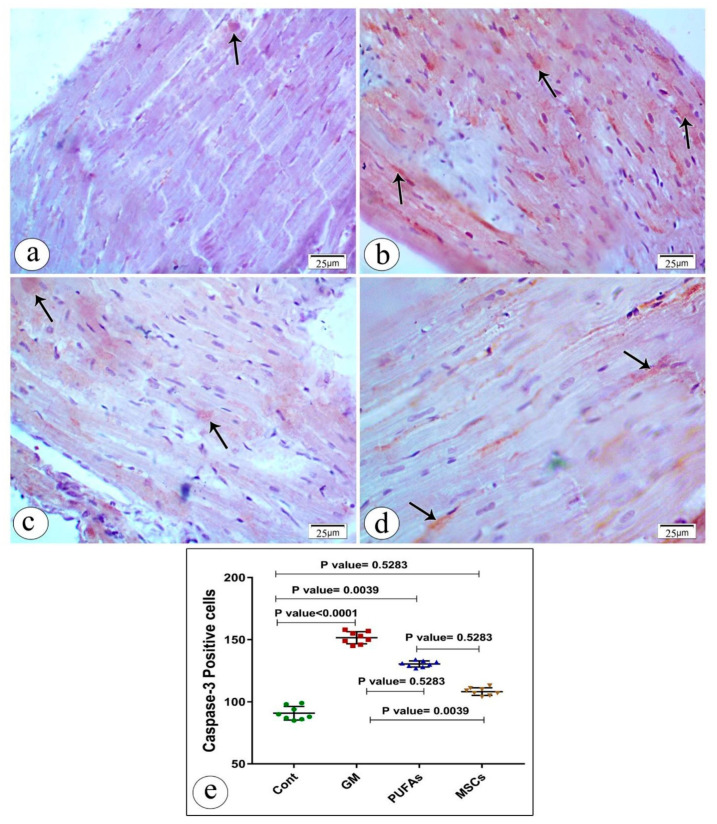
Immunostained sections of heart tissue stained with caspase-3 antibody: (**a**) control group showing mild positive caspase-3 immunostaining in the sarcoplasm of the muscle fibers (arrow); (**b**) gentamycin-treated group showing highly positive caspase-3 immunostaining in the sarcoplasm of the muscle fibers(arrow); (**c**) gentamycin- and PUFA-treated group showing moderate positive caspase-3 immunostaining in the sarcoplasm section in the cardiac muscle of control group(arrow); (**d**) gentamycin- and MSC-treated group showing moderate to mild positive caspase-3 expression with lower caspase-3 expression than the gentamycin group(arrow). (**e**) Statistical analysis of caspase intensity in cardiac myocytes in all groups studied. Nonparametric statistics were used. Data were analyzed using the Kruskal–Wallis test followed by Dunn’s test. Results are expressed as medians and interquartile ranges. Differences were considered significant at *p* < 0.05.

**Figure 7 pharmaceutics-14-01322-f007:**
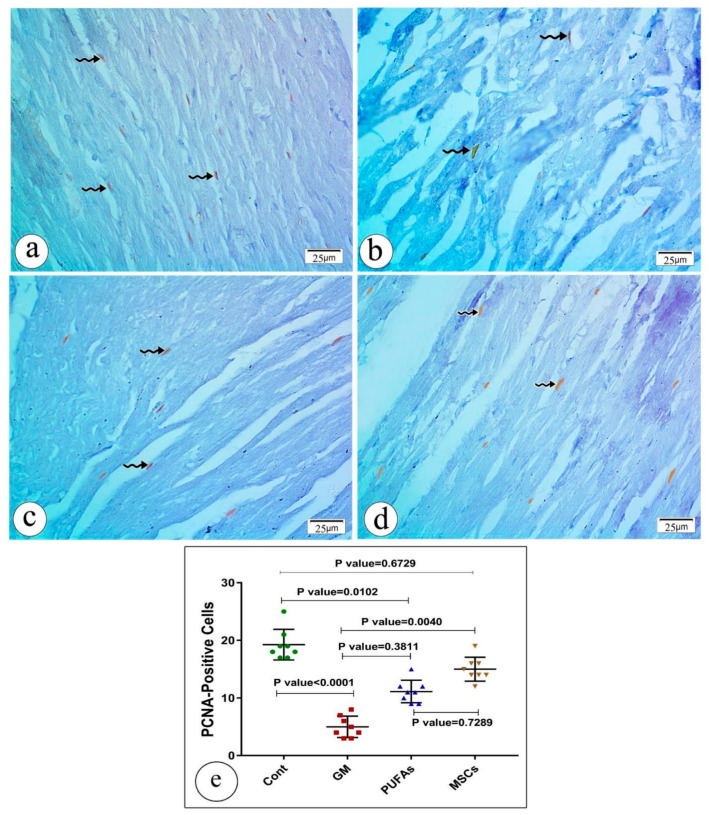
Immunostained sections of heart tissue PCNA expression in cardiac myocytes in all studied groups. Positive staining is shown by a dark-brown color: (**a**) control group showing moderately positive PCNA immunoreactivity in some myocytes (wavy arrow); (**b**) gentamycin-treated group showing almost absent PCNA immunoreactivity expression (wavy arrow); (**c**) gentamycin- and PUFA-treated group showing moderately positive immunoreactivity (wavy arrow); (**d**) gentamycin- and MSC-treated group showing moderately positive PCNA immunoreactivity almost similar to the control (wavy arrow). (**e**) Statistical analysis of PCNA expression in cardiac myocytes in all groups studied. Nonparametric statistics were used. Data were analyzed using the Kruskal–Wallis test followed by Dunn’s test. Results are expressed medians and interquartile ranges. Differences were considered significant at *p* < 0.05.

**Figure 8 pharmaceutics-14-01322-f008:**
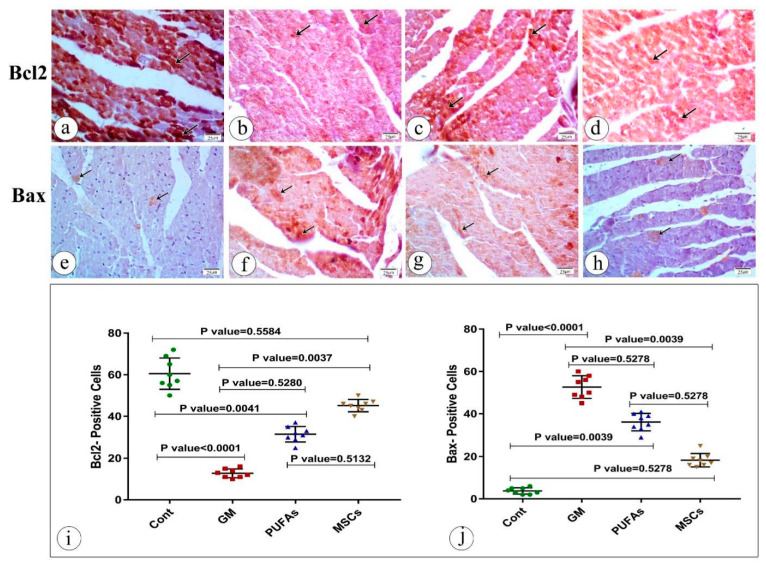
Immunostained sections of heart tissue of Bcl2 and Bax expression in cardiac myocytes in all studied groups. Positive staining is shown by a brown or red color: (**a**) control group showing strong Bcl2 immunohistochemical staining of control heart tissue in most of cardiomyocytes (arrow); (**b**) gentamycin-treated group showing minor positive Bcl2 immunoreactivity in cardiomyocytes (arrow); (**c**) gentamycin- and PUFA-treated group showing moderately positive Bcl2 immunoreactivity in some of the cardiomyocytes (arrow); (**d**) gentamycin- and MSC-treated group showing strong positive Bcl2 immunoreactivity in most cardiomyocytes (arrow); (**e**) control group showing positive Bax immunohistochemical staining of control heart tissue in a few myocytes (arrow); (**f**) gentamycin-treated group showing strong positive Bax immunoreactivity (overexpression of Bax) in the cardiomyocytes (arrow); (**g**) gentamycin- and PUFA-treated group showing moderately positive Bax immunoreactivity in some of the cardiomyocytes (arrow); (**h**) gentamycin- and MSC-treated group showing slight dark-red positive Bax immunoreactivity in most myocytes almost similar to the control (arrow). (**i**,**j**) respectively showing Statistical analysis of Bcl2 and Bax expression in cardiac myocytes in all groups studied. Nonparametric statistics were used. Data were analyzed using the Kruskal–Wallis test followed by Dunn’s test. Results are expressed as medians and interquartile ranges. Differences were considered significant at *p* < 0.05.

**Figure 9 pharmaceutics-14-01322-f009:**
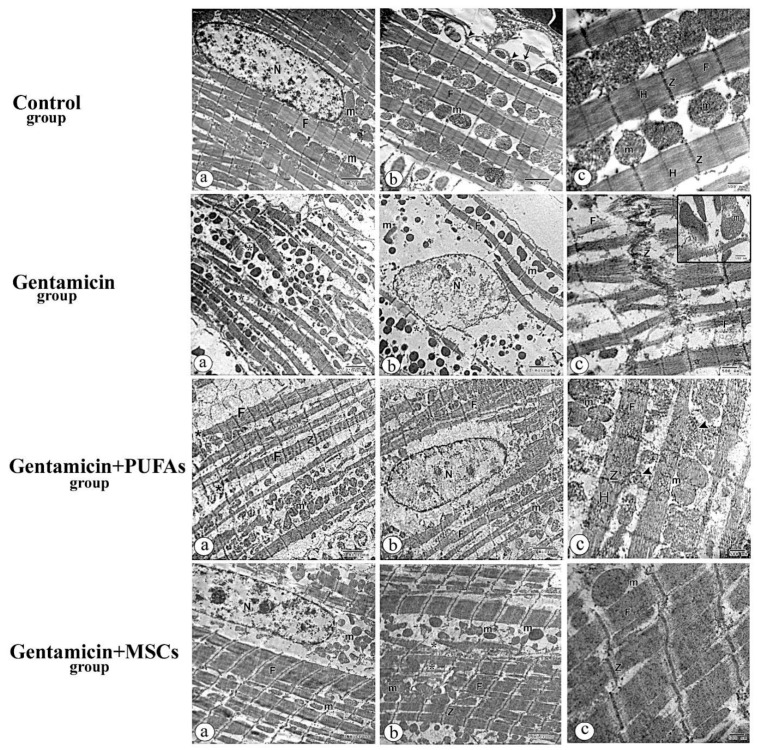
Electron micrographs of heart sections from rats of the control group: (**a**) centrally located nucleus (N), regular arrangement of the myofibrils (F), and mitochondria (m); (**b**) normal cell membrane of cardiac myocytes (arrow) with invaginations at T-tubules (arrow heads); (**c**) Z lines (Z) and H lines (H). Electron micrographs of heart sections from rats of the gentamycin-treated group: (**a**,**b**) separation of myofibrils with increased interfibrillar spaces (star) and myofibrillar fragmentation and degeneration (F) and irregular nucleus (N); (**c**) discontinuation and disarrangement of Z lines (Z), with disorganization of mitochondria (m in the incet); Electron micrographs of heart sections from rats of the gentamycin- and PUFA-treated group: (**a**) normal arrangement of myofibrils (F) with slight loss of myofibrils (star); (**b**) normal appearance of nucleus (N); (**c**) well-arranged mitochondria with some disorganization (arrow head) and continuation of Z lines (Z) and H lines (H) in some areas but not others (arrow). Electron micrographs of heart sections from rats of the gentamycin- and MSC-treated group: (**a**) normal arrangement of myofibrils (F) with a nearly normal nucleus (N); (**b**) mitochondria with a normal appearance regained (M); (**c**) continuation of Z lines (Z).

**Table 1 pharmaceutics-14-01322-t001:** Identity, sources, and working dilution of antibodies used in immunophenotyping studies.

Target	Primary Antibody Supplier	CAT. NO	Dilution	Incubation	Isotype	Secondary Antibody
CD90	Mouse monoclonal IgG1 κ (Santa Cruz biotech, CA, USA)	sc-53116	1 µg per 1 × 10^6^ cells	30 min	Normal mouse IgG1 (Alexa Fluor^®^ 647) conjugated isotype control immunoglobulin from mouse sc-24636	Goat anti-mouse IgG (H + L) recombinant secondary antibody, Alexa Fluor™ Plus 647 Waltham, MA, USA, A55060 Incubation time (30 min)
CD105	Mouse monoclonal antibody (Santa Cruz biotech, CA, USA)	sc-20072	1 µg per 1 × 10^6^ cells	30 min	Normal mouse IgG1 (Alexa Fluor^®^ 647) conjugated isotype control immunoglobulin from mouse sc-24636
CD29	Mouse monoclonal IgG1 κ (Santa Cruz biotech, CA, USA)	sc-9970	1 µg per 1 × 10^6^ cells	30 min	Normal mouse IgG1 (Alexa Fluor^®^ 647) conjugated isotype control immunoglobulin from mouse sc-24636
CD44	Mouse monoclonal antibody (Santa Cruz biotech, CA, USA)	sc-7297	1 µg per 1 × 106 cells	30 min	Normal mouse IgG1 (Alexa Fluor® 647) conjugated isotype control immunoglobulin from mouse sc-24636
CD45	Mouse monoclonal IgG1 κ (Santa Cruz biotech, CA, USA)	sc-1178	1 µg per 1 × 10^6^ cells	30 min	Normal mouse IgG1 (Alexa Fluor^®^ 647) conjugated isotype control immunoglobulin from mouse sc-24636
CD34	Mouse monoclonal IgG1 κ (Santa Cruz biotech, CA, USA	sc-7324	1 µg per 1 × 10^6^ cells	30 min	Normal mouse IgG1 (Alexa Fluor^®^ 647) conjugated isotype control immunoglobulin from mouse sc-24636

**Table 2 pharmaceutics-14-01322-t002:** Identity, sources, and working dilution of antibodies used in immunohistochemistry studies.

Target	Primary Antibody Supplier	CAT. NO	Dilution	Incubation	Antigen Retrieval	Secondary Antibody
Caspase-3	Rabbit polyclonal antibody, (AB clonal Technology, Woburn, MA, USA)	A0214	1:100	30 min	Boiling in citrate buffer (pH 6.0), 15 min	(ScyTek) Incubation time (30 min)
PCNA	Mouse monoclonal antibody (Novus Biologicals, Centennial, CO, USA)	NB500-106SS	1:100	30 min	Boiling in citrate buffer (pH 6.0), 15 min
BAX	Rabbit polyclonal antibody (Biospes, Chongqing, China)	YPA2175	1:100	60 min	Boiling in citrate buffer (pH 6.0), 15 min
BCL2	Rabbit polyclonal Anti- BCL2 antibody (Biospes, Chongqing, China)	YPA2275	1:100	60 min	Boiling in citrate buffer (pH 6.0), 15 min

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
