# Peer review of "Therapeutic Potential of Mesenchymal Stem Cells versus Omega n − 3 Polyunsaturated Fatty Acids on Gentamicin-Induced Cardiac Degeneration"

_pharmaceutics, 2022, doi:10.3390/pharmaceutics14071322_

Round 1

Reviewer 1 Report

The manuscript is presenting data about the toxic effect of gentamicin on the mouse heart and protecting effect of PUFAs and MSCs. The idea is interesting but there are many inaccuracies, not properly explained or proved parts, the English language should be strongly checked. The manuscript needs a strong revision and correction.

  1. The Introduction part and particularly section about the stem cells is written quite superficially. The authors are talking generally about the stem cells as to the not scientific audience. There is nothing written about the MSCs and their heart regenerating possibilities, compared to the other types of stem cells.
  2. The authors’ choice to cultivate MSCs with 100 μm L-ascorbic acid 2-phosphate is very doubtful, since this compound is used to differentiate cells. There are data that 50 uM and higher concentrations of L-ascorbic acid 2-phosphate strongly affects the BM-MSCs and their differentiation potential (J of Biosc and Bioeng, vol.105, No.6, 586-594, 2008). The proper controls of intracellular calcium or osteogenic differentiation background should be estimated.
  3. The methodological parts describing the MSCs immunophenotyping and differentiation are also written superficially, i.e. no detail explanation about the used antibodies and staining conditions, no composition of differentiation induction media, no explanation what staining colors means and how to interpret the data. Methodological part should be exact and clear as much as possible; it is not enough to redirect to some citations. Description of used antibodies in immunohistological and other methodological parts should include catalog No. The similar remarks are for the other method parts.
  4. Result part. The immunophenotyping of MSCs by showing only two positive markers is usually not enough.
  5. In the introduction part, the authors were talking about the SOD, glutathione peroxidase and catalase, but in the result part the only catalase left. Why the catalase was chosen? Catalase is neutralizing only peroxide, but not other ROS.
  6. The light microscopy micrographs (Fig. 3-5) in the result part are also quite superficially explained. It is necessary to explain in details what changes in gentamicin group compared to the control group are, what do they mean and what the MSCs and PUFAs do. It is not enough to write or to show “the oval nucleus” as in Fig.3. Moreover, the nucleus pointed in the control cells and gentamicin group look the same.

The text is explaining muscle striation but the arrows are pointing to the cardiomyocyte nucleus (Fig. 3). The manuscript also says that “necrosis, fibrosis, edema, and angiogenesis were evaluated“ but there are no explanations what changes mean what and how they should be interpreted.

  1. The placing of Figures near the text that explains the figures would improve the understand of the results. The legends should explain all changes or shortcuts that are in the Figures. The mice groups better to name as “control group”, gentamicin group” ant etc.
  2. The fig. 4. There is no explanation what color means what and how to interpret the changes.   5. The same remarks as for Fig. 3 and Fig. 4.
  3. The quantification of color intensity of histological and immunohistological sections by Image J programme is very doubtful. This programme needs to select areas manually and to evaluate size of the objects and then to evaluate these objects such as cells, dots or something else. However, the evaluation of histological sections according to the color intensity all over the section is hardly possible. The way of calculation is not explained.
  4. The caspase 3 evaluation according to the intensity of brawn color is also very doubtful and not explained in the method part. In parallel to the immunohistological measurements (caspase3, collagen, PCNA and other) the gene expression or quantitative measurement such as ELISA should be done. To state about the effects just from the qualitative immunocytochemistry staining, particularly when the staining is doubtful, is not enough. Such is Fig.7, which needs PGR or other more exact measurements, since it is impossible to state about significant changes by showing two or three stained nuclei.
  5. The electron microscopy micrographs are most informative and the best in this manuscript. The histological and immunohistological micrographs need an additional and more quantitative measurements, and much better explanation.
  6. Since the authors are talking much about the functions and morphology of mitochondria, the measurements of Bax/Bcl2 or caspase 9would prove that apoptosis is mitochondria-related process. The caspase 3 does not prove that the apoptosis is mitochondria-related process.
  7. The MScs and PUFAs were used as separate protectors that have no interconnection. The pharmacological question what would happen if they would be added together remains not answered.
  8. The discussion should be more structured and concrete pointing on the main findings of this manuscript in comparison with other authors. Discussion should not include parts or statements that are not proved by the results. For example, 377 line “There is production of nitric oxide (NO) associated with GM“ and many other.
  9. The English language should be strongly checked. There are unclear sentences such as line 40 “The activities of SOD, GSH-Px, catalase and 40 respiratory components (cytochrome c, NADH) [9].“ and other; the “Mesenchymal stem cells“ in the middle of sentence does not require a capital letter and so on.

Author Response

The manuscript is presenting data about the toxic effect of gentamicin on the mouse heart and protecting effect of PUFAs and MSCs. The idea is interesting but there are many inaccuracies, not properly explained or proved parts, the English language should be strongly checked. The manuscript needs a strong revision and correction.

We thank you for your kind words about our manuscript, Actually your comments are very valuable for us and by responding to them, our manuscript became more better

  1. The Introduction part and particularly section about the stem cells is written quite superficially. The authors are talking generally about the stem cells as to the not scientific audience. There is nothing written about the MSCs and their heart regenerating possibilities, compared to the other types of stem cells.

   Author responsea paragraph was added in the introduction discussing the MSCs and    their heart regenerating possibilities, compared to the other types of stem cells. Page 4

  1. The authors’ choice to cultivate MSCs with 100 μm L-ascorbic acid 2-phosphate is very doubtful, since this compound is used to differentiate cells. There are data that 50 uM and higher concentrations of L-ascorbic acid 2-phosphate strongly affects the BM-MSCs and their differentiation potential (J of Biosc and Bioeng, vol.105, No.6, 586-594, 2008). The proper controls of intracellular calcium or osteogenic differentiation background should be estimated.

Author response:  The method of cultivation of isolated MSCs was rewritten in page 5

  1. The methodological parts describing the MSCs immunophenotyping and differentiation are also written superficially, i.e. no detail explanation about the used antibodies and staining conditions, no composition of differentiation induction media, no explanation what staining colors means and how to interpret the data. Methodological part should be exact and clear as much as possible; it is not enough to redirect to some citations.

Author response

The procedure was written in the main steps to be understandable without any confusion and was referenced. We provided details for the used antibodies as sources and Catalog numbers. Also, we inserted details for differentiation media with catalog numbers in the process of differentiation. The stains used were supplied with more detailed components. The procedures and usage of the media and stains were performed according to the instructions of the manufacturers.

  1. Description of used antibodies in immunohistological and other methodological parts should include catalog No. The similar remarks are for the other method parts.

Author response The catalog numbers were added in the text and in a table

  1. Result part. The immunophenotyping of MSCs by showing only two positive markers is usually not enough.

Author response Two other positive markers were added

  1. In the introduction part, the authors were talking about the SOD, glutathione peroxidase and catalase, but in the result part the only catalase left. Why the catalase was chosen? Catalase is neutralizing only peroxide, but not other ROS.

Author response

In the assay of oxidant/antioxidant balance we chose MDA as an indicator for lipid peroxidation and catalase as an indicator for antioxidant defense. The role of peroxide is highly involved in many cardiovascular, neurological, and hepatic diseases. Our resources allowed us to perform these tests to permit the completion of other estimations and investigations. In addition, the role of catalase is comparable to and essential as other enzymes.

  1. The light microscopy micrographs (Fig. 3-5) in the result part are also quite superficially explained. It is necessary to explain in details what changes in gentamicin group compared to the control group are, what do they mean and what the MSCs and PUFAs do. It is not enough to write or to show “the oval nucleus” as in Fig.3. Moreover, the nucleus pointed in the control cells and gentamicin group look the same.

Author response The detailed were added in the legends

The text is explaining muscle striation but the arrows are pointing to the cardiomyocyte nucleus (Fig. 3). The manuscript also says that “necrosis, fibrosis, edema, and angiogenesis were evaluated“ but there are no explanations what changes mean what and how they should be interpreted.

  1. The placing of Figures near the text that explains the figures would improve the understand of the results. The legends should explain all changes or shortcuts that are in the Figures. The mice groups better to name as “control group”, gentamicin group” ant etc.

   Author response The legends explained all changes that are in the figures. The rat groups was named as “control group”, gentamicin group” ant etc.

  1. The fig. 4. There is no explanation what color means what and how to interpret the changes.   5. The same remarks as for Fig. 3 and Fig. 4.

Author response :The legends were rewritten to explain the findings in the figures

  1. The quantification of color intensity of histological and immunohistological sections by Image J programme is very doubtful. This programme needs to select areas manually and to evaluate size of the objects and then to evaluate these objects such as cells, dots or something else. However, the evaluation of histological sections according to the color intensity all over the section is hardly possible. The way of calculation is not explained.

Author response Immunohistochemistry (IHC) is a potent method that uses the specific binding of an antibody to an antigen to detect and localize particular antigens in cells and tissue and frequently used supplementary testing method in morphological and surgical pathology for cell classification and diagnosis [Taylor, 2014]. In numerous cancers, including those of the breast, gastrointestinal tract, lung, hematolymphoid, and central nervous systems, the use of IHC has recently extended to examine predictive and prognostic biomarkers [Chen ZE, Lin ., 2015& Yong etal 2014] The College of American Pathologists has set guidelines for the standardization and analytic validation of immunohistochemistry testing [ Lin and Chen .,2014& . Fitzgibbons) etal., 2014].

The automated methods for high volume processing with reproducibility, IHC has become an essential ancillary technique in clinical diagnostics in anatomic pathology [Magaki etal., 2019]

immunohistochemistry (IHC) is a useful method for biomarker validation. it enables direct observation of biomarker expression in histologically significant sections of the tissue under investigation. This is a significant benefit over solubilized  materials for biochemical examination, which might result in false negative findings if there are just a few biomarker-positive cells in a backdrop of biomarker-negative tissue components  so validated IHC tests may be quickly introduced into clinical practice[Rizzardi. Etal., 2012]

In the context of histology specimens, digital image analysis, also known as tissue image analysis or quantitative image analysis, refers to the automated extraction of data from digitized histological sections using analytical algorithms. There are several methods and algorithm techniques to choose from, and the quantity and functionality of applications looks to be rapidly expanding [Chen etal., 2017]. Standard applications for automated cell or nuclear detection are available. Other software packages even have incorporated simplified machine learning algorithms that allow users to create custom image analysis solutions that can identify and quantify wider tissue properties, such as automated necrotic tissue recognition.   Deep learning in medical imaging has lately been the subject of comprehensive reviews. Aeffner etal 2018

In the current work The quantitative analysis of CD117, proliferating cell nuclear antigen (PCNA),) was performed by using Image J software (Version 1.53i). The DAB signal was quantified using image J software to estimate the differences in immunoreactivity for PCNA. From “Analyze menu,” choose “Set Measurement” and check “Area,” “Max. gray value,” and “Mean gray value” from the resulting pop-up window. Many circles were drawn within the cytoplasm of the cells and were measured. The results were copied to excel file, and the optical density was calculated according to the following equation: OD = log (Max. gray intensity/mean gray intensity) to obtain the degree of immunoreactivity (darkness) of the stained cells by the DAB signal.[ Abd-Elhafeez

 Etal., 2021]

 Taylor CR (2014) Immunohistochemistry in surgical pathology: Principles and practice In: Day CE (ed) Histopathology: methods and protocols. Methods in molecular biology Springer, New York, pp 81–110

 Chen ZE, Lin F (2015) Overview of predictive biomarkers and integration of IHC into molecular pathology In: Fan L, Jeffrey P (eds) Handbook of practical immunohistochemistry: frequently asked questions, 2nd edn. Springer, New York

  Yong WH, Dry SM, Shabihkhani M (2014) A practical approach to clinical and research biobanking. Methods Mol Biol 1180:137–162 [PubMed: 25015146]

Lin F, Chen Z (2014) Standardization of diagnostic immunohistochemistry: literature review and Geisinger experience. Arch Pathol Lab Med 138:1564–1577 [PubMed: 25427038

 Fitzgibbons PL, Bradley LA, Fatheree LA et al. (2014) Principles of analytic validation of immunohistochemical assays: guideline from the college of american pathologists pathology and laboratory quality center. Arch Pathol Lab Med 138:1432–1443 [PubMed: 24646069]

c S, Hojat SA, Wei B, So A, Yong WH. An Introduction to the Performance of Immunohistochemistry. Methods Mol Biol. 2019;1897:289-298. doi: 10.1007/978-1-4939-8935-5_25. PMID: 30539453; PMCID: PMC6749998.

Rizzardi AE, Johnson AT, Vogel RI, et al. Quantitative comparison of immunohistochemical staining measured by digital image analysis versus pathologist visual scoring. Diagn Pathol. 2012;7:42. Published 2012 Jun 20. doi:10.1186/1746-1596-7-42

Chen JM, Li Y, Xu J, et al. Computer-aided prognosis on breast cancer with hematoxylin and eosin histopathology images: A review. Tumour Biol. 2017;39(3):1010428317694550.

Aeffner F, Adissu HA, Boyle MC, et al. Digital Microscopy, Image Analysis, and Virtual Slide Repository. ILAR J. 2018;59(1):66-79. doi:10.1093/ilar/ily007

Abd-Elhafeez HH, Soliman SA, Attaai AH, Abdel-Hakeem SS, El-Sayed AM, Abou-Elhamd AS. Endocrine, Stemness, Proliferative, and Proteolytic Properties of Alarm Cells in Ruby-Red-Fin Shark (Rainbow Shark), Epalzeorhynchos frenatum (Teleostei: Cyprinidae). Microsc Microanal. 2021 Aug 4:1-14. doi: 10.1017/S1431927621012265. Epub ahead of print. PMID: 34344492.

  1. The caspase 3 evaluation according to the intensity of brawn color is also very doubtful and not explained in the method part. In parallel to the immunohistological measurements (caspase3, collagen, PCNA and other) the gene expression or quantitative measurement such as ELISA should be done. To state about the effects just from the qualitative immunocytochemistry staining, particularly when the staining is doubtful, is not enough. Such is Fig.7, which needs PGR or other more exact measurements, since it is impossible to state about significant changes by showing two or three stained nuclei.

Author response Unfortunately, we have shortage of primers necessary for gene expression/ ELISA kits, and time is not in our side to get them in the short time allowable for revision submission.  If we ordered primers from outside Egypt, we must wait 3 months to get them.  And according to our basic knowledge, immunohistochemical staining or western blotting target to identify if there is an increase in the expression of a certain protein in a certain tissue. So, we could only manage to do IMH studies.

  1. The electron microscopy micrographs are most informative and the best in this manuscript. The histological and immunohistological micrographs need an additional and more quantitative measurements, and much better explanation.

 Author response:  More explanations were added to the micrographs to be better

  1. Since the authors are talking much about the functions and morphology of mitochondria, the measurements of Bax/Bcl2 or caspase 9would prove that apoptosis is mitochondria-related process. The caspase 3 does not prove that the apoptosis is mitochondria-related process.

Author response Immunostaining  of Bax and Bcl2  were done and the results were added. And we referred to them in the introduction; pages 2-3 and in discussion,  pages 14-15.

  1. The MScs and PUFAs were used as separate protectors that have no interconnection. The pharmacological question what would happen if they would be added together remains not answered.

Author response The MScs and PUFAs were used as separate protectors that have no interconnection. The pharmacological question what would happen if they would be added together remains not answered. Our results showed that both MSCs and PUFAs treatments has ameliorated gentamicin-induced cardiac degeneration. Their actions were mediated through upregulating PCNA and BCL2 expression, down-regulating caspase 3, BAX  expression. Also, treatment with either MSCs or PUFAs has improved cardiac antioxidant activity and reduced oxidative stress.  Consequently, if they would be added together, they would act synergistically and would induce more pronounced cardiac protective effects.

  1. The discussion should be more structured and concrete pointing on the main findings of this manuscript in comparison with other authors. Discussion should not include parts or statements that are not proved by the results. For example, 377 line “There is production of nitric oxide (NO) associated with GM“ and many other.

Author response: The discussion was more structured and concrete pointing on the main findings of this manuscript in comparison with other authors

  1. The English language should be strongly checked. There are unclear sentences such as line 40 “The activities of SOD, GSH-Px, catalase and 40 respiratory components (cytochrome c, NADH) [9].“ and other; the “Mesenchymal stem cells“ in the middle of sentence does not require a capital letter and so on.

Author response The English language was checked by the The Egyptian Knowledge Bank (EKB) and the certificate was attached.

Reviewer 2 Report

Here, the authors attempted to examine whether mesenchymal stem cells (MSCs) and dietary n-3 polyunsaturated fatty acids (PUFAs) contribute to cardiac regeneration after gentamicin-induced cardiac injury. Unfortunately, the paper needs an extensive English language copyediting and the data are poorly presented, hence requiring a major revision before being considered for publication. The manuscript should be better resubmitted de novo as several figures need to be redrawn and even reconsidered to be sufficiently informative. In addition, a number of results are borderline at best and do not provide sufficient evidence for the conclusions made by the authors. The paper relies on histology and immunohistochemistry while requiring at least Western blot or RT-qPCR of the tissue lysate. Semi-quantitative analysis of immunohistochemical images is the complementary technique, and the conclusions cannot be based solely on this approach (especially when counting arbitrary units of percent of the positively stained area rather than percent of positive cells). Below are the specific comments:

Abstract

Please indicate the number of rats in each group and decipher IP route (intraperitoneally).

Introduction

Please unify the terminology and use MSCs instead of SCs. Please also describe the sources of MSCs as this is not clear in the current manuscript.

Materials and Methods

Please indicate the catalogue numbers and suppliers for each reagent used.

Please use only non-parametric statistics and replace mean and standard deviation with median and interquartile range, as you have applied Kruskal-Wallis test and Mann-Whitney U-test. Please also indicate the adjustment for multiple comparisons (e.g. Dunn's test or false discovery rate).

Results

Please replace all bar graphs with univariate scatterplots (GraphPad Prism: box-and-whiskers, min to max, show all points). Please also replace the asterisks and letters with exact P values indicated in a numerical manner. Make the graphs coloured and indicate both of the comparisons with GM group (i.e., MSC and PUFA treatments) in all figures even if they are not significant. Please also replace the group numbers (I-IV) with exact treatments as it is more convenient for the readers. 

Please make the Figure 1 more attractive; currently, it is not well shaped.

Figure 2: replace MG with GM (gentamicin).

Figure 3 is nonsensical as the sections are vastly different from each other simply because of sectioning artifacts. Intergroup differences are negligible here and this figure must be redrawn or excluded from the manuscript.

Figure 4 also suffers from tears which likely occurred during sectioning and the images must be replaced. In addition, they are not well representative and do not correspond to the graph below. Please indicate the collagen area (%) at the Y axis.

Figure 5 is also not informative. Please indicate what it means in the figure legend.

Figure 6: what was the measure of caspase intensity? If there were arbitrary units, it must be indicated at the Y axis. The same issue relates to the Figure 7 (PCNA expression).

Figures 8-11 must be merged into one figure as they reflect different experimental groups rather than distinct experiments.

Discussion

The discussion is very convoluted and should be substantially abridged.

Author Response

We thank you for your kind words about our manuscript, Actually your comments are very valuable for us and by responding to them, our manuscript became more better

Here, the authors attempted to examine whether mesenchymal stem cells (MSCs) and dietary n-3 polyunsaturated fatty acids (PUFAs) contribute to cardiac regeneration after gentamicin-induced cardiac injury. Unfortunately, the paper needs an extensive English language copyediting and the data are poorly presented, hence requiring a major revision before being considered for publication. The manuscript should be better resubmitted de novo as several figures need to be redrawn and even reconsidered to be sufficiently informative. In addition, a number of results are borderline at best and do not provide sufficient evidence for the conclusions made by the authors. The paper relies on histology and immunohistochemistry while requiring at least Western blot or RT-qPCR of the tissue lysate. Semi-quantitative analysis of immunohistochemical images is the complementary technique, and the conclusions cannot be based solely on this approach (especially when counting arbitrary units of percent of the positively stained area rather than percent of positive cells). Below are the specific comments:

Author response The English language was checked by the The Egyptian Knowledge Bank (EKB) and the certificate was attached.

Author response Unfortunately, we have shortage of primers necessary for gene expression   and time is not in our side to get them in the short time allowable for revision submission. If we ordered primers from outside Egypt, we must wait 3 months to get them.  And according to our basic knowledge, immunohistochemical staining or  western blotting target to identify if there is an increase in the expression of a certain protein in a certain tissue. So, we could only manage to do IMH studies.

Abstract

Please indicate the number of rats in each group and decipher IP route (intraperitoneally).

Author response

The number of each group was added in the abstract.

IP  was written (intraperitoneally)

Introduction

Please unify the terminology and use MSCs instead of SCs. Please also describe the sources of MSCs as this is not clear in the current manuscript.

Author response

The terminology was unified

The source of MSCs was well described

Materials and Methods

Please indicate the catalogue numbers and suppliers for each reagent used.

Author response

The catalog numbers and the suppliers for each reagents were added

Please use only non-parametric statistics and replace mean and standard deviation with median and interquartile range, as you have applied Kruskal-Wallis test and Mann-Whitney U-test. Please also indicate the adjustment for multiple comparisons (e.g. Dunn's test or false discovery rate).

Author response:

Non-parametric statistics was used and mean and standard deviation  was replaced with median and interquartile range.

Results

Please replace all bar graphs with univariate scatterplots (GraphPad Prism: box-and-whiskers, min to max, show all points). Please also replace the asterisks and letters with exact P values indicated in a numerical manner. Make the graphs coloured and indicate both of the comparisons with GM group (i.e., MSC and PUFA treatments) in all figures even if they are not significant. Please also replace the group numbers (I-IV) with exact treatments as it is more convenient for the readers.

Author response:

  • All bar graphs were replaced with univariate scatterplots (GraphPad Prism: box-and-whiskers, min to max, show all points.
  • The asterisks and letters were replaced with exact P values indicated in a numerical manner.
  • The graphs are colored and indicate both comparisons with GM group (i.e., MSC and PUFA treatments) in all figures even if they are not significant.
  • The group numbers (I-IV) were replaced with exact treatments

Please make the Figure 1 more attractive; currently, it is not well shaped.

Author response We did our best to make the figure more attractive and well-shaped

Figure 2: replace MG with GM (gentamicin).

Author response : MG  was replaced  with GM (gentamicin).

Figure 3 is nonsensical as the sections are vastly different from each other simply because of sectioning artifacts. Intergroup differences are negligible here and this figure must be redrawn or excluded from the manuscript.

Author response: we tried to make the figure more sensical and tried to write the detailed explanation in figure legend.

Figure 4 also suffers from tears which likely occurred during sectioning and the images must be replaced. In addition, they are not well representative and do not correspond to the graph below. Please indicate the collagen area (%) at the Y axis.

Author response: The images were changed.  the collagen area (%) was added on at the Y axis of the graph. Photos descriping the collagen contents only were added.

Figure 5 is also not informative. Please indicate what it means in the figure legend.

Author response: The figure legend was rewritten to explain the figure

Figure 6: what was the measure of caspase intensity? If there were arbitrary units, it must be indicated at the Y axis. The same issue relates to the Figure 7 (PCNA expression).

Author response: The photos were changed and counting was done

Figures 8-11 must be merged into one figure as they reflect different experimental groups rather than distinct experiments.

Author response :The figures 8-11 were merged in one figure

Discussion

The discussion is very convoluted and should be substantially abridged.

Author response: The discussion became more structured and concrete pointing on the main findings of this manuscript in comparison with other authors

Reviewer 3 Report

The topic submitted is novel and adds significant research data to the existing field of Mesenchymal Stem Cells versus Omega n-3 polyunsaturated fatty acids research. The article is not very well articulated and needs English language revisions and even formatting of the manuscript as per the MDPI guidelines. The manuscript needs to be checked for statistical significance, especially in fig 2. The introduction needs to be concise. Abstract and conclusion should include a sentence proposing the future direction of the present research. Also the commercial aspects of how pharma or vaccine industries can benefit.  Most of the methodology lacks citing the references from where the technique was adopted and used to conduct the study. 

Author Response

The topic submitted is novel and adds significant research data to the existing field of Mesenchymal Stem Cells versus Omega n-3 polyunsaturated fatty acids research.

Author responseWe thank you very much for your kind comment. Actually your comments are very valuable for us and by responding to them, our manuscript became more better

The article is not very well articulated and needs English language revisions and even formatting of the manuscript as per the MDPI guidelines. The manuscript needs to be checked for statistical significance, especially in fig 2.

Author response

  • The English language was checked by the The Egyptian Knowledge Bank (EKB) and the certificate was attached.
  • The manuscript was formatted according to the MDPI guidelines

The introduction needs to be concise.

Author response: The introduction was rewritten according to the comments of the reviewers. We tried to make it more concise

Abstract and conclusion should include a sentence proposing the future direction of the present research. Also the commercial aspects of how pharma or vaccine industries can benefit. 

Author response: PUFAs and MSCs exert ameliorative effects against gentamicin-induced cardiac degeneration. They can protect the myocardium by promoting differentiation of myocardial cells, increasing apoptosis resistance, and inhibiting oxidative stress which are ideal qualities for cardiovascular repair. However, our findings showed that MSCs had better efficacy compared to PUFAs. This will provide an insight into consideration of MSCs therapy as a novel and promising approach for the treatment of cardiovascular diseases.

 Most of the methodology lacks citing the references from where the technique was adopted and used to conduct the study.

Author response :We tried to add the missed references

Round 2

Reviewer 1 Report

The manuscript has been improved, some new experiments and calculations were added. The text also has been improved. However, there are still some inaccuracies.

1.       It is agreed that the term „expression“ is used only for the genes, while for the proteins it is „level“. This should be corrected all over the manuscript. Already in the abstract the sentence “BCL2 and Bax expression….” is confusing, since it is not clear what the authors are taking about, i.e. genes or proteins.

2.       The explanation of group II in the abstract (line 12-13) is in the square brackets, while other in bend brackets.

3.      Typing inaccuracies: Line 123 – the abbreviation „Bm-MScs“ ; line 144 – „....BM-MSCs, Cells...“; line 249 ” pyknotic. interstitial…”, some words are in bold font (line251 and 257); line 422 “PUVA and MSC groups.”

4.       Sometimes the „Bax“ (line 196) and sometimes the „BAX“ (line 185) is written. The same inaccuracies is with “BCL2” and “Bcl2”. It should be corrected and written alike all over the manuscript.

5.      What about the external apoptotic pathway? Was it measured by the authors?

6.      If the authors use the abbreviation "BM-MSCs", it should be the same all over the manuscript. Now, sometimes BM-MSCs, sometimes only MSCs were used that is not clear are the cells the same.

Author Response

Comments and Suggestions for Authors

The manuscript has been improved, some new experiments and calculations were added. The text also has been improved. However, there are still some inaccuracies.

Author response: We thank you very much for your kind words. Actually, your valuable comments made our manuscript  more better

  1. It is agreed that the term „expression“ is used only for the genes, while for the proteins it is „level“. This should be corrected all over the manuscript. Already in the abstract the sentence “BCL2 and Bax expression….” is confusing, since it is not clear what the authors are taking about, i.e. genes or proteins.

Author response: The term expression is changed into level all over the manuscript

  1. The explanation of group II in the abstract (line 12-13) is in the square brackets, while       other in bend brackets.

Author response:  All brackets are now in bend type

  1. Typing inaccuracies: Line 123 – the abbreviation „Bm-MScs“ ; line 144 – „....BM-MSCs, Cells...“; line 249 ” pyknotic. interstitial…”, some words are in bold font (line251 and 257); line 422 “PUVA and MSC groups.”

Author response: All mentioned inaccuracies are corrected

  1. Sometimes the „Bax“ (line 196) and sometimes the „BAX“ (line 185) is written. The same inaccuracies is with “BCL2” and “Bcl2”. It should be corrected and written alike all over the manuscript.

Author response: Bcl2 and Bax were used all over the manuscript

  1. What about the external apoptotic pathway? Was it measured by the authors?

Author response: From our knowledge, both pathways eventually stimulate the caspase cascade and end up with the same terminal.  So we measured caspase in our manuscript.

  1. If the authors use the abbreviation "BM-MSCs", it should be the same all over the manuscript. Now, sometimes BM-MSCs, sometimes only MSCs were used that is not clear are the cells the same.

Author response: The term BM-MSCs was used all over the manuscript

Reviewer 2 Report

The authors well addressed all my comments and significantly revised the paper, which can be now accepted for publication.

Author Response

Actually, we thank you very much for your comments that make our manuscript better

And we thank you for accepting our manuscript after the modifications that we done according to your valuable comments.  
